# DJMix: Unsupervised Task-agnostic Augmentation for Improving Robustness

## Abstract

Convolutional Neural Networks (CNNs) are vulnerable to unseen noise on input images at the test time, and thus improving the robustness is crucial. In this paper, we propose `DJMix`, a data augmentation method to improve the robustness by mixing each training image and its discretized one. Discretization is done in an unsupervised manner by an autoencoder, and the mixed images are nearly impossible to distinguish from the original images. Therefore, `DJMix` can easily be adapted to various image recognition tasks. We verify the effectiveness of our method using classification, semantic segmentation, and detection using clean and noisy test images.

## 1 Introduction

CNNs are the de facto standard components of image recognition tasks and achieve excellent performance. However, CNNs are vulnerable to unseen noise on input images. Such harmful noise includes not only adversarially generated noise (Szegedy et al., 2014; Goodfellow et al., 2014), but also naturally possible noise such as blur by defocusing and artifacts generated by JPEG compression (Vasiljevic et al., 2016; Hendrycks & Dietterich, 2019). Natural noise on input images is inevitable in the real world; therefore, making CNNs robust to natural noise is crucial for practitioners.

A simple approach to solving this problem is adding noise to training images, but this does not make models generalize to unseen corruptions and perturbations (Vasiljevic et al., 2016; Geirhos et al., 2018; Gu et al., 2019). For example, even if Gaussian noise of a certain variance is added during training, models fail to generalize to Gaussian noise of other variances. Nonetheless, some data augmentation methods are effective for improving robustness. For example, Yin et al. reported that extensive augmentation, such as `AutoAugment` (Cubuk et al., 2019), improves the robustness. Similarly, Hendrycks et al. proposed to mix differently augmented images during training to circumvent the vulnerability. We will further review previous approaches in Section 2.

Despite the effectiveness of these data augmentation and mixing approaches, these methods require handcrafted image transformations, such as rotation and solarizing. Particularly when geometrical transformations are used, the mixed images cannot have trivial targets in non classification tasks, for instance, semantic segmentation and detection. This lack of applicability to other tasks motivates us to introduce robust data augmentation without such transformations.

In this paper, we propose Discretizing and Joint Mixing (`DJMix`) which mixes original and discretized training images to improve the robustness. The difference between the original and obtained images is nearly imperceptible, as shown in Figure 1, which enables the use of `DJMix` in various image recognition tasks. In Section 3, we will introduce `DJMix` and analyze it empirically and theoretically. We show that `DJMix` reduces mutual information between inputs and internal representations to ignore harmful features and improve CNNs' resilience to test-time noise.

To benchmark the robustness of CNNs to unseen noise, Hendrycks & Dietterich (2019) introduced `ImageNet-C` as a corrupted counterpart of the `ImageNet` validation set (Russakovsky et al., 2015). CNN models are evaluated using this dataset on the noisy validation set, whereas they are trained without any prior information on the corruptions on the original training set. Similarly, Geirhos et al. created noisy `ImageNet` and compared different behaviors between humans and CNN models with image noise. In addition to these datasets designed for classification, we cre-

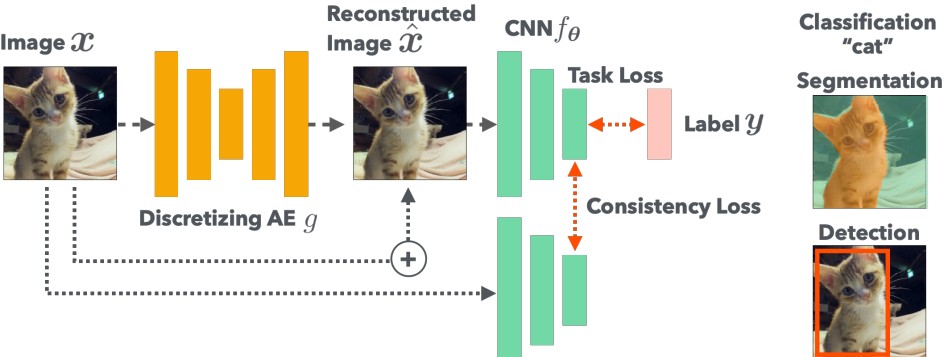

Figure 1: Schematic view of `DJMix`. In `DJMix`, CNN models are trained to minimize divergence between the features of each input image $f_{\boldsymbol{\theta}}(\boldsymbol{x})$ and its discretized image $f_{\boldsymbol{\theta}}(\hat{\boldsymbol{x}})$, as well as the task-specific loss between $f_{\boldsymbol{\theta}}(\hat{\boldsymbol{x}})$ and the label $\boldsymbol{y}$. This pipeline barely affects the appearance of input images, and thus, can be used for various image recognition tasks, *e.g.*, classification, (semantic) segmentation, and detection.

ated `Segmentation-C` and `Detection-C` datasets, which are corrupted counterparts of the `PASCAL-VOC` validation sets (Everingham et al., 2015).

We demonstrate the robustness of CNN models trained with `DJMix` on various tasks using these benchmark datasets in Section 4. Additionally, we perform experimental analyses, including ablation studies, to verify `DJMix` in Section 5. In summary, our contributions are summarized as follows:

1. We introduce `DJMix`, a simple task-agnostic data augmentation method for improving robustness. `DJMix` mixes the original and discretized training images and can be straightforwardly adapted to various image recognition tasks. We empirically demonstrate the effectiveness of this approach.

2. We analyze `DJMix` theoretically from the Information Bottleneck perspective, which could help analyze other robust methods. We also investigate `DJMix` from the Fourier sensitivity perspective.

3. We create datasets, `Segmentation-C` and `Detection-C`, to benchmark the robustness of CNN models on semantic segmentation and detection tasks.

## 2 RELATED WORK

Small corruptions or perturbations on images can drastically change the predictions of CNN models. While adversarially generated noise, *i.e.*, adversarial examples (Szegedy et al., 2014; Goodfellow et al., 2014), can be thought of as the worst case, natural noise also harms the performance of CNNs. Such natural noise includes blurs and artifacts generated by JPEG compression (Vasiljevic et al., 2016; Hendrycks & Dietterich, 2019). Because of this vulnerability, CNN models sometimes predict inconsistently among adjacent video frames (Gu et al., 2019; Hendrycks & Dietterich, 2019). For the real-world application of CNNs, this vulnerability needs to be overcome. A strong defense against adversarial examples is adversarial training, where CNN models are trained with adversarial examples (Goodfellow et al., 2014; Madry et al., 2018). Unfortunately, this approach fails in natural noise, because CNNs trained on a specific type of noise do not generalize to other types of noise (Geirhos et al., 2018; Vasiljevic et al., 2016). Instead, we need robust methods that are agnostic to test-time noise *a priori* (Hendrycks & Dietterich, 2019).

Data augmentation is a practical approach to improving the generalization ability on clean data, *e.g.*, by randomly flipping and cropping (Krizhevsky et al., 2012; He et al., 2016), mixing different images (Zhang et al., 2018; Tokozume et al., 2018), or erasing random regions (DeVries & Taylor; Zhong et al., 2020). Some types of data augmentation are also reported to improve robustness. For example, strong data augmentation, namely, `AutoAugment` (Cubuk et al., 2019), can improve the

robustness (Yin et al., 2019). Similarly, `AugMix` is a data augmentation method to alleviate the problem by mixing images that are differently augmented from each input image. CNNs exploit the texture or higher-frequency domains of images (Jo & Bengio, 2017; Ilyas et al., 2019; Wang et al., 2020), and thus, CNNs trained on detextured `ImageNet` images by style transfer show robustness to noise on input images (Geirhos et al., 2019).

Orthogonal to manipulating input images, enhancing CNN architectures or components is also a possible direction. Larger CNN models with feature aggregation used in `DenseNet` (Huang et al., 2017) and `ResNeXt` (Xie et al., 2017) show better robustness to natural noise (Gu et al., 2019; Hendrycks & Dietterich, 2019). `MaxBlur` has been proposed to improve the shift invariance of subsampling operations used in pooling operations, *e.g.*, MaxPooling, and enhance the robust performance (Zhang, 2019).

Our approach, `DJMix`, belongs to the first ones, which use data augmentation to enhance robustness. Unlike previous methods, `DJMix` applies imperceptible and task-agnostic augmentation to images. This property allows us to use `DJMix` for various image recognition tasks.

## 3   `DJMix` FOR ROBUST DATA AUGMENTATION

A CNN model $f_{\boldsymbol{\theta}} : \mathbb{R}^D \to \mathbb{R}^{D'}$ is usually trained to minimize the task loss $\ell(f_{\boldsymbol{\theta}}(\boldsymbol{x}), \boldsymbol{y})$, where $\boldsymbol{x} \in \mathbb{R}^D$ is an input image, and $\boldsymbol{y} \in \mathbb{R}^{D'}$ is its target. When the task is a $D'$-category classification task, $\boldsymbol{y}$ is a one-hot vector and $\ell$ is cross-entropy.

`DJMix` uses a pair of loss functions, the task loss $\ell(f_{\boldsymbol{\theta}}(\hat{\boldsymbol{x}}), \boldsymbol{y})$ and the consistency loss $d(f_\theta(\hat{\boldsymbol{x}}), f_\theta(\boldsymbol{x}))$. Then, CNN models are trained to minimize

$$\ell(f_{\boldsymbol{\theta}}(\hat{\boldsymbol{x}}), \boldsymbol{y}) + \gamma d(f_{\boldsymbol{\theta}}(\hat{\boldsymbol{x}}), f_{\boldsymbol{\theta}}(\boldsymbol{x})), \tag{1}$$

where $\gamma$ is a positive coefficient. $\hat{\boldsymbol{x}} \in \mathbb{R}^D$ is a discretized image of an input image $\boldsymbol{x}$, which we will describe in Section 3.1, and $d$ is a divergence, such as the Jensen–Shannon divergence (Section 3.2). We will discuss why `DJMix` improves robustness in Sections 3.3 and 3.4, both theoretically and empirically.

### 3.1   DISCRETIZATION OF IMAGES

`DJMix` discretizes each input image $\boldsymbol{x}$ into $g(\boldsymbol{x})$, where $g : \mathbb{R}^D \to \mathbb{R}^D$ is a discretizing autoencoder (DAE), whose bottleneck is discretized. Specifically, we used the Vector-Quantized Variational AutoEncoder used by van den Oord et al. (2017) and Razavi et al. (2019). This DAE $g$ has a bottleneck of dimension $C$ and discretizes the features by vector quantization with the codebook size of $2^K$. DAE is pretrained on training data to minimize $\mathbb{E}_{\boldsymbol{x}}\|g(\boldsymbol{x}) - \boldsymbol{x}\|_2^2$ in an unsupervised manner.

As we will show in Section 5, mixing each input image and its discretized one improves the robustness. More precisely, instead of using $\hat{\boldsymbol{x}} = g(\boldsymbol{x})$, we use

$$\hat{\boldsymbol{x}} = \beta\boldsymbol{x} + (1 - \beta)g(\boldsymbol{x}), \tag{2}$$

where $\beta \in [0, 1]$ is sampled from a random distribution. Following Zhang et al. (2018), we adopt Beta distribution. Although this mixing strategy is similar to that of `AugMix`, some differences exist. A major difference is the discrepancy between $\boldsymbol{x}$ and $\hat{\boldsymbol{x}}$. Because `AugMix` applies geometric and color-enhancing operations to obtain $\hat{\boldsymbol{x}}$, its appearance is different from $\boldsymbol{x}$, whereas `DJMix` yields a nearly identical $\hat{\boldsymbol{x}}$ from $\boldsymbol{x}$. A minor difference is the task loss: `DJMix` uses $\ell(f_{\boldsymbol{\theta}}(\hat{\boldsymbol{x}}), \boldsymbol{y})$, whereas `AugMix` uses $\ell(f_{\boldsymbol{\theta}}(\boldsymbol{x}), \boldsymbol{y})$. We will analyze this difference in Section 5.

### 3.2   CONSISTENCY LOSS

The consistency loss $d(f_\theta(\hat{\boldsymbol{x}}), f_\theta(\boldsymbol{x}))$ forces a CNN model to map $\boldsymbol{x}$ and $\hat{\boldsymbol{x}}$ closely and make these representations indistinguishable. Following Hendrycks et al. (2020), we use the Jensen–Shannon

(JS) divergence as the divergence $d$. We compare other divergences and distances with the JS divergence in Section 5.

`DJMix` appears similar to `AugMix` (Hendrycks et al., 2020), as both use the mixing of images and the consistency loss. However, the details of the mixing processes are different: whereas `AugMix` yields a different looking $\hat{x}$ from $x$, `DJMix` augments a similar $\hat{x}$ to $x$. Owing to this property, `DJMix` can be used in various image recognition tasks, as we will show in Section 4. Additionally, the task loss of `DJMix` uses mixed images as $\ell(f_{\boldsymbol{\theta}}(\hat{x}), \boldsymbol{y})$, whereas that of `AugMix` uses the original image as $\ell(f_{\boldsymbol{\theta}}(x), \boldsymbol{y})$. Empirically, we found that $\ell(f_{\boldsymbol{\theta}}(\hat{x}), \boldsymbol{y})$ improves the robustness compared with $\ell(f_{\boldsymbol{\theta}}(x), \boldsymbol{y})$, which we will show in Section 5.1.

### 3.3 FROM INFORMATION BOTTLENECK PERSPECTIVE

The Information Bottleneck objective (Tishby et al., 2000) can be written with an intermediate feature $z$ of a model $f_{\boldsymbol{\theta}}$ as $\max_{\boldsymbol{\theta}} \mathcal{I}(z, \boldsymbol{y}; \boldsymbol{\theta})$ s.t. $\mathcal{I}(x, z; \boldsymbol{\theta}) \leq I$, where $\mathcal{I}(w, v) := D_{\mathrm{KL}}(p(w, v) \| p(w)p(v))$ is mutual information between $w$ and $v$, and $I$ is a positive constraint. Supervised training is expected to maximize $\mathcal{I}(z, \boldsymbol{y}; \boldsymbol{\theta})$. However, without the constraint, $z$ highly likely contains unnecessary details of the input; then the models learn vulnerable representation (Alemi et al., 2017; Fisher & Alemi, 2020). Importantly, `DJMix` introduces this constraint to improve the robustness by ignoring task-irrelevant details. For the following theorem, we assume that $\beta$ in Equation (2) is 0, and $f_{\boldsymbol{\theta}}$ and $g$ have enough capacities to achieve training losses being 0.

**Theorem 1.** *Let $z$ be $f_{\boldsymbol{\theta}}(x)$. After convergence of the model $f_{\boldsymbol{\theta}}$ trained with DJMix, mutual information is constrained by the logarithm of the codebook size,* i.e.*,*

$$\mathcal{I}(x, z; \boldsymbol{\theta}) \leq K. \tag{3}$$

*Proof.* After convergence, $d(f_{\boldsymbol{\theta}}(\hat{x}), f_{\theta}(x))$ becomes 0, or equivalently, $f_{\boldsymbol{\theta}}(\hat{x}) = f_{\boldsymbol{\theta}}(x)$ from the assumption. $x$ is quantized into a codeword $\hat{x} \in \{1, 2, \ldots, 2^K\}$ in the DAE $g$. Therefore, we obtain

$$
\begin{aligned}
K &= \mathcal{H}(\mathrm{Uniform}\{1, 2, \ldots, 2^K\}) \quad (\mathcal{H}: \text{entropy}) \\
&= \mathcal{H}(\hat{x}) \\
&\geq \mathcal{H}(\hat{x}) - \mathcal{H}(\hat{x} \mid f_{\boldsymbol{\theta}}(\hat{x})) \\
&= \mathcal{I}(f_{\boldsymbol{\theta}}(\hat{x}), \hat{x}) \\
&\geq \mathcal{I}(f_{\boldsymbol{\theta}}(x), x) \qquad \text{(from Data Processing Inequality)} \\
&= \mathcal{I}(x, z)
\end{aligned}
$$

$\square$

### 3.4 FROM FOURIER SENSITIVITY PERSPECTIVE

Figure 2 presents the sensitivity of CNN models trained with and without `DJMix` to additive noise of Fourier-basis vectors (Yin et al., 2019). Here, we used `WideResNet` trained on `CIFAR10`. As can be seen, `DJMix` improves robustness to a wide range of frequencies: from lower frequencies, depicted in the center area, to higher frequencies, which appearing in the edges. These results imply why CNN models trained with `DJMix` show robustness to input images with noise. The experiments discussed in Section 4 further demonstrate more empirical robustness of `DJMix`.

## 4 EXPERIMENTS AND RESULTS

In this section, we present experimental results. We first introduce experimental settings and new datasets, `Segmentation-C`, and `Detection-C`, whose input images are artificially corrupted to measure the robustness. Then, we present empirical results and comparisons with other methods in Section 4.1 for classification, Section 4.2 for semantic segmentation, and Section 4.3 for detection. We conducted the experiments three times with different random seeds for each setting and reported the averaged values, except for `ImageNet` experiments. We describe the additional details of the experiments in Appendix B.

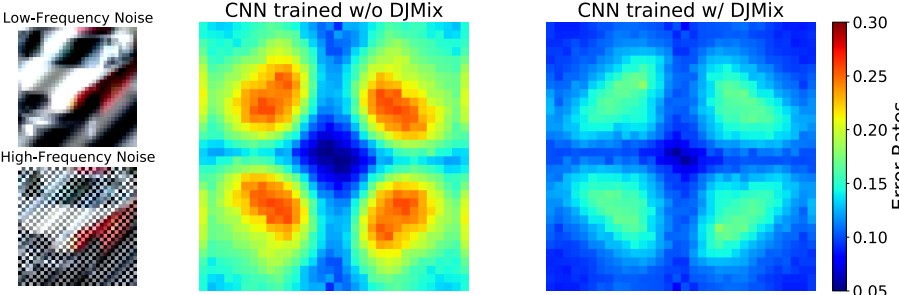

Figure 2: `DJMix` reduces the sensitivity of CNN models to additive noise. Fourier-basis vectors whose $L_2$ norm is 4 are added to 2,048 randomly sampled test samples of `CIFAR10` whose image size is $32 \times 32$, and pixels present the averaged error rates. Each pixel corresponds to a certain Fourier basis, *e.g.*, the center area of lower frequencies and the edges of higher frequencies. The leftmost images are examples of images with additive noise enhanced for visualization.

IMPLEMENTATION

We implemented `DJMix` as well as CNN models using `PyTorch` (Paszke et al., 2019) and used `FAISS` (Johnson et al., 2017) to make the nearest neighbor search in DAE faster. We used classification models used by Hendrycks et al. (2020) [1]. For segmentation and detection tasks, we used `DeepLab-v3` and `Faster-RCNN` from `torchvision`[2], whose backbone networks are `ResNet-50` (He et al., 2016) pretrained on `ImageNet` (Russakovsky et al., 2015).

DAE is pretrained on each dataset for the classification task and on `ImageNet` for other tasks. We set a dictionary size to 512, *i.e.*, $K = 9$, following Razavi et al. (2019). We set the parameters of Beta distribution $(\beta_0, \beta_1)$ for mixing in Equation (2) to $(1.0, 0.5)$, and the coefficient for the consistency loss $\gamma$ to 1.0.

`DJMix` is a task-agnostic method and can improve robustness by itself. Additionally, `DJMix` can be incorporated with task-specific data augmentation. We introduce a `DJMix` variant that applies random data augmentation (`DJMix`+RA), consisting of `AugMix`'s augmentation operations. We describe more details of RA in Appendix B.5.

DATASETS

For classification, we used three datasets, `CIFAR10`, `CIFAR100` (Krizhevsky, 2009), and `ImageNet` (Russakovsky et al., 2015), consisting of 10, 100, and 1,000 categories, respectively. We trained CNN models on clean training sets and evaluated the models using the accompanying clean test sets. We also evaluated the models on corrupted test sets (`CIFAR10-C`, `CIFAR100-C`, and `ImageNet-C`) proposed by Hendrycks & Dietterich (2019), which are created to measure the behavior of CNN models with 15 common corruptions.

To benchmark the robustness in segmentation and detection tasks, we created `Segmentation-C` and `Detection-C` datasets from `PASCAL VOC-2012` (Everingham et al., 2015). Namely, we degenerated images of the test set of `PASCAL VOC-2012` for segmentation and the validation set of `PASCAL VOC-2012` for detection using 10 degeneration operations used in `ImageNet-C`: `gaussian_noise`, `shot_noise`, `impulse_noise`, `snow`, `frost`, `fog`, `brightness`, `contrast`, `pixelate`, and `jpeg_compression`. We omitted five blur operations, namely `defocus_blur`, `glass_blur`, `motion_blur`, `zoom_blur`, and `gaussian_blur`, because the expected outputs for segmentation and detection under corruption are not trivial. Examples of `Segmentation-C` are presented in Figure 3. Following the convention, we trained models on the augmented dataset of `PASCAL-VOC` (Hariharan et al., 2011) for segmentation and the union of train and validation datasets of `VOC-2007` and `VOC-2012` for detection. Similar to

---

[1] `https://github.com/google-research/augmix`
[2] `https://github.com/pytorch/vision/tree/v0.7.0`

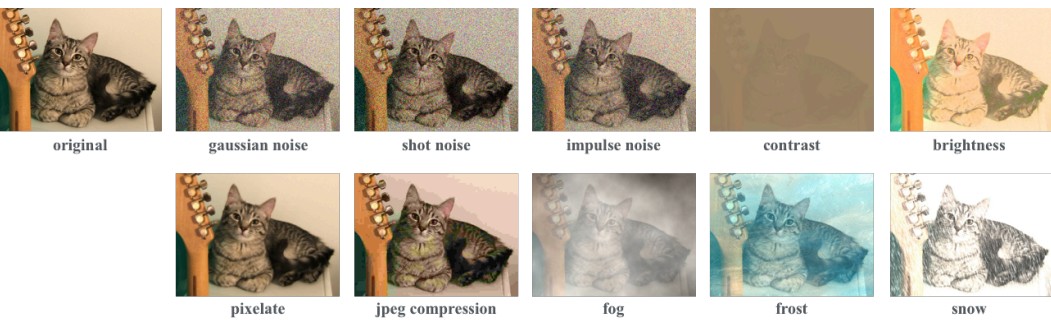

Figure 3: Examples of the `Segmentation-C` dataset with the severest noise of each corruption type. We created this dataset to benchmark the robustness of segmentation networks by degenerating validation images with 10 operations, such as Gaussian noise. Similarly, we created `Detection-C` as a detection counterpart of `Segmentation-C`.

`Detection-C`, Michaelis et al. (2019) also created corrupted test images of detection datasets for autonomous driving.

METRICS

For the classification task, we use the error rates as the metric on the original test sets. On the corrupted data, we measure the corrupted error rate $E_{c,s}$, where $c$ is the corruption type, *e.g.*, Gaussian noise, and $s$ is the severity level, and report the statistics. Precisely, we use the averaged scores $\mathbb{E}_{c,s}E_{c,s}$ for `CIFAR10-C` and `CIFAR100-C`, and Corruption Error $\mathbb{E}_s E_{c,s}/\mathbb{E}_s E_{c,s}^{\text{AlexNet}}$ for `ImageNet-C`, following Hendrycks & Dietterich (2019), where $E_{c,s}^{\text{AlexNet}}$ is the error rate of `AlexNet` (Krizhevsky et al., 2012). For the segmentation task, we report the mean intersection over union (mIoU) on the clean data. On `Segmentation-C`, we use corrupted mIoU $I_{c,s}$ and report averaged mIoU $\mathbb{E}_s I_{c,s}$. Similarly, for the detection task, we report mean average precision (mAP) on the clean data and averaged corrupted mAP $\mathbb{E}_s A_{c,s}$ of corrupted mAP $A_{c,s}$ on `Detection-C`.

4.1 CLASSIFICATION

We trained models on `CIFAR10`, `CIFAR100`, and `ImageNet`. For `CIFAR10` and `CIFAR100`, we used `DenseNet-BC` ($k = 12$, $d = 100$) (Huang et al., 2017), `WideResNet-28-2` (Zagoruyko & Komodakis, 2016), and `ResNeXt-29` (Xie et al., 2017). For `ImageNet`, we used `ResNet-50` (He et al., 2016).

The results on `CIFAR10` and `CIFAR100` are presented in Table 1 with comparison with the baseline, `mixup` (Zhang et al., 2018), `AugMix`, and `AugMix` without geometric transformations (GT). Table 1 shows that `AugMix` without GTs degenerates performance both on clean and corrupted data from `AugMix`, indicating that the robustness of `AugMix` heavily depends on GTs. `DJMix` shows balanced performance between clean and corrupted data, without such GTs. `DJMix` with RA further decreases error rates on corrupted datasets, as well as on clean datasets. We present the results on `ImageNet` in Table 2. Again, `DJMix` decreases Corruption Errors, particularly when strong data augmentation is introduced (`DJMix+RA`).

4.2 SEMANTIC SEGMENTATION

We trained `DeepLab-v3` (Chen et al., 2017) on `PASCAL-VOC`. The logits before upsampling are used for the consistency loss.

Table 3 shows the mIoU comparison of baseline, `AugMix` without GTs (`AugMix*`), `DJMix`, and `DJMix` with random augmentation without GTs (`DJMix+RA*`). We omitted GTs, because defining the targets for mixed images that differently applied GTs are not trivial for segmentation and detection tasks. `AugMix` w/o GT uses pairs of original and augmented images, *i.e.*, the width is set to 2. As can be seen, `DJMix` improves the robustness, especially when combined with extra data augmentation. In some cases, such as Gaussian noise, shot noise, and impulse noise, `DJMix+RA` markedly

Table 1: Test error rates (↓) comparison on clean test sets (`CIFAR10` and `CIFAR100`) and corrupted sets (`CIFAR10-C` and `CIFAR100-C`). AugMix w/o GT is a variant of `AugMix` without using GTs. `DJMix+RA` is a variant of `DJMix` with random data augmentation.

|  |  | Baseline | mixup | AugMix | AugMix w/o GT | DJMix | DJMix+RA |
|---|---|---|---|---|---|---|---|
| CIFAR10 | WideResNet | 4.65 | 4.45 | 5.07 | 6.14 | 5.02 | 4.74 |
|  | DenseNet | 5.12 | 5.03 | 5.55 | 6.54 | 5.45 | 4.98 |
|  | ResNeXt | 4.13 | 3.66 | 4.17 | 5.42 | 4.82 | 4.31 |
| CIFAR10-C | WideResNet | 29.6 | 23.4 | 11.1 | 18.4 | 15.7 | 11.5 |
|  | DenseNet | 32.7 | 27.3 | 12.1 | 19.0 | 17.6 | 12.9 |
|  | ResNeXt | 30.6 | 26.1 | 10.7 | 18.4 | 14.4 | 11.1 |
| CIFAR100 | WideResNet | 24.3 | 24.2 | 25.6 | 27.6 | 25.7 | 24.4 |
|  | DenseNet | 25.4 | 25.4 | 26.3 | 27.8 | 26.2 | 24.7 |
|  | ResNeXt | 22.0 | 21.0 | 22.3 | 24.5 | 25.1 | 23.2 |
| CIFAR100-C | WideResNet | 56.4 | 51.6 | 36.4 | 46.0 | 44.3 | 38.3 |
|  | DenseNet | 59.3 | 54.4 | 37.7 | 47.3 | 47.2 | 39.9 |
|  | ResNeXt | 56.2 | 50.8 | 34.1 | 44.4 | 42.1 | 35.9 |

Table 2: Test error rates (↓) on `ImageNet` and Corruption Error (↓) on `ImageNet-C` using `ResNet-50`. `DJMix` works on `ImageNet` and improves the robustness.

|  | Clean | Gauss. | Shot | Impulse | Defocus | Glass | Motion | Zoom | Snow | Frost | Fog | Bright | Contrast | Pixel | JPEG | Average |
|---|---|---|---|---|---|---|---|---|---|---|---|---|---|---|---|---|
| Baseline | **23.2** | 76.0 | 77.7 | 78.8 | 78.1 | 88.9 | 80.7 | 81.0 | 80.4 | 78.4 | 70.7 | 61.7 | 73.9 | 72.5 | 76.3 | 76.8 |
| DJMix | 23.6 | 72.6 | 74.4 | 74.7 | 76.1 | 86.2 | 79.9 | 80.8 | 77.0 | 76.0 | 68.6 | 60.9 | 70.2 | 71.1 | 76.1 | 74.6 |
| DJMix+RA | 23.8 | **66.8** | **67.5** | **68.9** | **71.6** | **85.1** | **73.8** | **73.1** | **73.2** | **72.1** | **62.4** | **59.2** | **67.8** | **70.6** | **73.9** | **70.4** |

enhances the performance from `DJMix` and `AugMix` w/o GT, which implies the importance of the combination of task-specific and task-agnostic augmentation in practice.

### 4.3 DETECTION

We trained `Faster-RCNN` (Ren et al., 2015) on `PASCAL-VOC`. The consistency loss between the output logits of the backbone network is used for training. Table 4 shows that `DJMix` yields better performance on almost all corruption types. As semantic segmentation, we compare baseline, `AugMix`*, `DJMix`, and `DJMix+RA`*. Similarly to semantic segmentation, `DJMix` markedly improves the robustness in the detection task.

## 5 ANALYSIS

### 5.1 ABLATION STUDIES

DESIGN OF TASK LOSS: The task loss of `DJMix` presented in Equation (1) is $\ell(f_{\boldsymbol{\theta}}(\hat{\boldsymbol{x}}), \boldsymbol{y})$, but $\ell(f_{\boldsymbol{\theta}}(\boldsymbol{x}), \boldsymbol{y})$, as `AugMix`, is also a possible choice. We compare these choices in Table 5 (a, b). $\ell(f_{\boldsymbol{\theta}}(\hat{\boldsymbol{x}}), \boldsymbol{y})$ improves the robustness compared with $\ell(f_{\boldsymbol{\theta}}(\boldsymbol{x}), \boldsymbol{y})$.

CHOICE OF CONSISTENCY LOSS: `DJMix` uses JS divergence as the consistency loss, but other divergences can also be used as the loss function. Here, we compare the performance when JS divergence is replaced with KL divergence and L2 distance. As can be seen from Table 5 (a, c), JS and KL show similar performance, whereas L2 shows performance degeneration on corrupted data.

EFFECT OF DISCRETIZATION: We verify the effect of the discretization of `DJMix` using DAE by substituting the standard autoencoder for DAE. Namely, we removed vector quantization modules of DAE and pretrained the AE on the training data to minimize the reconstruction error as DAE. Table 5 (a, d) shows that discretization improves CNNs' robustness as expected from the Information Bottleneck perspective presented in Section 3.3.

EFFECT OF MIXING: Table 5 (e) shows test error rates of `DJMix` without mixing, *i.e.*, $\beta = 0$ in Equation (2) where only discretized images are used. The results show that mixing is indispensable to retain the performance on clean data. We present further experiments on betas in Appendix B.

Table 3: `DJMix` improves the performance of semantic segmentation when input images are corrupted. We present mIoU (↑) on `PASCAL-VOC` (Clean) and `Segmentation-C` (the rest). * indicates that augmentation without geometric transformations is used.

| | Clean | Gauss | Shot | Impulse | Snow | Frost | Fog | Bright | Contrast | Pixel | JPEG | Average |
|---|---|---|---|---|---|---|---|---|---|---|---|---|
| Baseline | 73.3 | 29.5 | 30.6 | 26.5 | 22.2 | 36.4 | 58.8 | 65.4 | 39.8 | 58.6 | 58.8 | 42.7 |
| `AugMix`* | **74.7** | 39.0 | 40.3 | 37.6 | 21.8 | 38.6 | 63.1 | 68.3 | 46.7 | 60.9 | 62.9 | 47.9 |
| `DJMix` | 73.5 | 34.7 | 35.3 | 30.3 | **23.5** | 38.8 | 59.2 | 66.5 | 40.2 | 62.3 | 63.3 | 45.4 |
| `DJMix+RA`* | 74.4 | **46.9** | **47.7** | **45.3** | 22.6 | **41.9** | **63.6** | **69.7** | **46.9** | **64.6** | **66.3** | **51.6** |

Table 4: `DJMix` improves the performance of detection when input images are corrupted. We present mAP (↑) on `PASCAL-VOC` (Clean) and `Detection-C` (the rest). * indicates that augmentation without geometric transformations is used.

| | Clean | Gauss | Shot | Impulse | Snow | Frost | Fog | Bright | Contrast | Pixel | JPEG | Average |
|---|---|---|---|---|---|---|---|---|---|---|---|---|
| Baseline | 76.5 | 37.9 | 39.7 | 33.8 | 46.4 | 51.7 | 64.6 | 70.1 | 47.3 | 47.4 | 51.1 | 49.0 |
| `AugMix`* | **76.8** | 38.1 | 40.0 | 35.8 | 45.6 | 51.1 | 64.9 | 70.4 | 48.0 | 47.2 | 52.1 | 49.3 |
| `DJMix` | **76.8** | 40.8 | 43.1 | 37.5 | 45.8 | 51.9 | 64.8 | 71.1 | 47.6 | 46.2 | 47.6 | 49.6 |
| `DJMix+RA`* | 76.2 | **45.6** | **48.5** | **44.7** | **46.8** | **52.8** | **65.3** | **72.2** | **48.6** | **47.4** | **54.7** | **52.7** |

## 5.2 COMPUTATIONAL OVERHEAD OF `DJMix`

We find that the computational overhead by the DAE is negligible. However, the number of *forward passes* affects the training time. For instance, the standard training on `CIFAR10` using `WideResNet` for 200 epoch requires approximately 1 hour in our environment. `DJMix` with two forward passes per update takes about 2 hours, and `AugMix` with three forward passes per update takes about 3 hours. Importantly, `DJMix` does not modify the components of CNNs as `AugMix`; therefore, these methods do not affect the test-time speed, which is preferable for the real-world applications.

## 6 CONCLUSION

In this paper, we have proposed `DJMix`, a novel task-agnostic approach to make CNN models robust to test-time corruption. To achieve task-agnostic robustness, we have used an autoencoder with a discretization bottleneck. Unlike previous approaches, the image modification of `DJMix` does not affect the appearance of images, which is useful for non classification tasks. Experiments have shown that `DJMix` improves the robustness of CNN models to input noise in semantic segmentation and detection, in addition to classification. We have found that combining task-specific and task-agnostic augmentation methods further improves performance on noisy images. We hope that data augmentation for robustness, including `DJMix`, bridges research and the real-world practice of deep learning.

Table 5: Test error rates (average / standard deviation, ↓) on `CIFAR10` and `CIFAR10-C` with various ablation settings.

| | CIFAR10 | CIFAR10-C |
|---|---|---|
| (a) `DJMix` | $5.02 \pm 0.22$ | $15.7 \pm 0.2$ |
| (b) `DJMix` w/ $\ell(f_\theta(\boldsymbol{x}), \boldsymbol{y})$ | $4.54 \pm 0.14$ | $17.8 \pm 0.5$ |
| (c) `DJMix` w/ KL | $4.88 \pm 0.13$ | $15.3 \pm 0.2$ |
|     `DJMix` w/ L2 | $4.70 \pm 0.25$ | $20.5 \pm 0.5$ |
| (d) `DJMix` w/o discretization | $4.44 \pm 0.09$ | $28.9 \pm 0.4$ |
| (e) `DJMix` w/o mixing | $7.71 \pm 0.19$ | $15.3 \pm 0.0$ |
| Baseline | $4.65 \pm 0.12$ | $29.6 \pm 0.3$ |

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

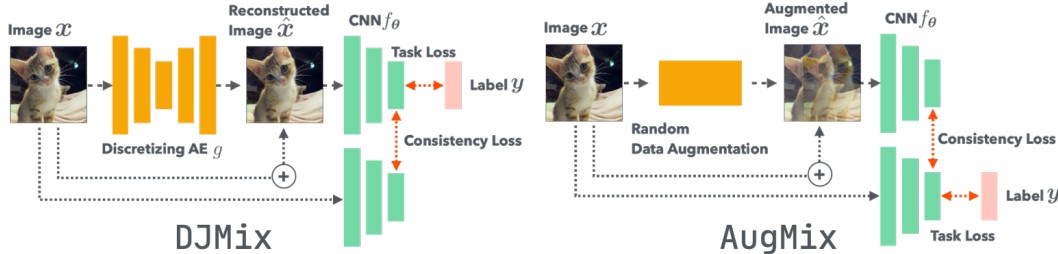

Figure 4: Schematic comparison of `DJMix` (ours) and `AugMix` of mixing two examples (Hendrycks et al., 2020). Both methods use mixing strategies, but the details are different. Notably, a reconstructed image of `DJMix` $\hat{x}$ is almost indistinguishable from the original image $x$.

## A    ADDITIONAL ABLATION STUDIES

### A.1    THE EFFECT OF BETA DISTRIBUTION PARAMETERS

Main experiments used $(\beta_0, \beta_1) = (1.0, 0.5)$ of Beta distribution for mixing $\hat{x} = \beta x + (1 - \beta)g(x)$, where $\beta \sim \text{Beta}(\beta_0, \beta_1)$. Figure 5 shows test error rates on different combinations of the parameters using `WideResNet`. Larger $\beta_0$ and smaller $\beta_1$ yield $\hat{x}$ close to $x$, and vice versa, which is reflected in the results on `CIFAR10`, *i.e.*, clean data. We used $(\beta_0, \beta_1) = (1.0, 0.5)$, which balances performance on clean and corrupted data.

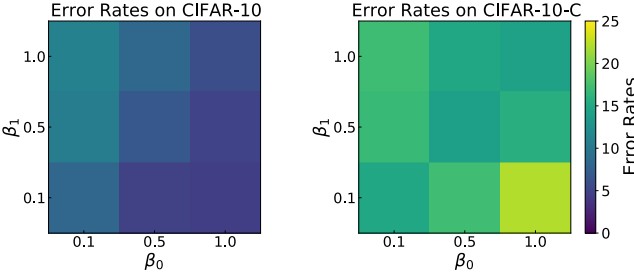

Figure 5: Test error rates on different combinations of $(\beta_0, \beta_1)$ of Beta distribution used for mixing. We used $(\beta_0, \beta_1) = (1.0, 0.5)$ in the main experiments.

## B    EXPERIMENTAL SETTINGS DETAILS

This section describes additional experimental settings.

### B.1    DISCRETIZING AUTOENCODER

We trained the autoencoder for 200 epochs to minimize the reconstruction error, and its codebook is updated by exponential moving average. The hyperparameters are identical to Razavi et al. (2019).

### B.2    CLASSIFICATION

We trained models on `CIFAR10`, `CIFAR100`, and `ImageNet`. For `CIFAR10` and `CIFAR100`, we used `DenseNet-BC` ($k = 12$, $d = 100$) (Huang et al., 2017), `WideResNet-28-2` (Zagoruyko & Komodakis, 2016) and `ResNeXt-29` (Xie et al., 2017). We trained these networks for 200 epochs using stochastic gradient descent with a momentum of 0.9 and setting an initial learning rate to 0.1 that decays by cosine annealing with warm restart (Loshchilov & Hutter, 2016). We set a weight decay to $1 \times 10^{-4}$ and a batch size to 128. We used data augmentation of random horizontal flipping, random cropping, and random erasing (Zhong et al., 2020) by default. For `ImageNet`, we

used `ResNet-50` (He et al., 2016) and trained it for 100 epochs using SGD with a momentum of 0.9 and a weight decay of $1 \times 10^{-4}$ . We set the batch size to 1,024 and an initial learning rate to 0.4 that decays at 30, 60, and 90th epochs. We used random cropping and horizontal flipping as the base data augmentation.

When training `ResNet-50` on `ImageNet`, we used automatic mixed precision (AMP) implemented in `PyTorch v1.6` to save the GPU memory consumption. We also used AMP for semantic segmentation and detection tasks.

## B.3 SEMANTIC SEGMENTATION

We trained `DeepLab-v3` (Chen et al., 2017) for 30 epochs with a batch size of 32 and a learning rate of $1.0 \times 10^{-3}$. We used SGD with a momentum of 0.9 and set its initial learning rate to 0.02. The learning rate is multiplied by a factor of $(1 - \frac{\text{iteration}}{\text{total iteration}})^{0.9}$ as Chen et al. (2017). See `https://github.com/pytorch/vision/tree/master/references/segmentation` for further details.

## B.4 DETECTION

We trained `Faster-RCNN` (Ren et al., 2015) for 26 epochs with a batch size of 32 and a learning rate of $1.0 \times 10^{-3}$. The learning rate is divided by 10 at 16th and 22nd epochs, while the first 1,000 iterations are the warmup period. See `https://github.com/pytorch/vision/tree/master/references/detection` for further details.

## B.5 RANDOM AUGMENTATION

We used random augmentation (RA) as task-specific data augmentation, which is orthogonal to `DJMix`. Basically, we followed the data augmentation module of `AugMix`, and the only difference is the *width*. Whereas `AugMix` sets the width to 3, `DJMix` uses the width of 1, *i.e.*, only a single stream of operations is applied to each input image. Each image augmented by RA is used as an input $x$ to `DJMix`.

