# OpenReview forum: "DJMix: Unsupervised Task-agnostic Augmentation for Improving Robustness"
_ICLR.cc/2021/Conference — Reject_

### Official Review · AnonReviewer2 · 2020-10-27
**Review of "DJMix: Unsupervised Task-agnostic Augmentation for Improving Robustness"**

**Rating:** 4
**Confidence:** 4

**Review:**

# Summary
This paper proposes an alternative data augmentation method for robust image classification. Training on augmented data has been shown to improve robustness. The proposed method mixes discretized images with real images and uses a consistency loss to enforce smoothness among those predictions. Experimental results include classification, detection and segmentation. Section 5 includes ablation studies on the choice of consistency loss and choice of discretization.

# Strong and Weak points

## Strong points

  * The empirical results include numbers of both classification, detection and segmentation. Thus showing the benefits of the proposed methods on a range of application areas.
  * Figure (3) shows quantifiable increase in robustness to perturbations based on fixed Fourier coefficients. Even though the phenomenon, as first described in Yin 2019, is not fully understood, this is an important step to make CNNs more robust.

## Weak points

  * In introduction, the argument is made that other methods (like AugMix and other augmentations) require hand crafted image transformations, such as rotation and solarizing. This paper proposes to replace the hand crafted image transformations with another neural network. However, it remains unclear how the known failure cases of neural networks (as described in the second paragraph of the introduction) do not also affect the auto encoder used for the image transformations.
  * As seen in Figure 2, a critical difference between DJMix and AugMix is the design of the Task loss. In AugMix, the task loss penalizes the prediction on clean data, in DJMix, the task loss penalizes the prediction on augmented data. However, I miss the reasoning, either theoretical or empirical, to justify this choice. How do the methods differ when using clean or augmented data for the task loss?
  * The baseline results do not match results published in literature. Table 1 notes that AugMix scores 36.4, for a WideResNet on CIFAR 100. However, the original paper, [1] reports 35.9 on the same experiment. I include the link to the paper below. Perhaps I am mistaking something here, but it’s curious that these results are not the same.
In either case of 35.9 or 36.4 being the correct number, both results are lower (better) than proposed method scores on the experiment, which scores the 38.3. I miss the reason why the proposed method scores a higher (worse) result on this experiment?
  * Table 2 makes a comparison against a weak baseline. The results are 76.8 for the baseline, and 70.4 for the proposed method. (Lower is better, so this would exemplify an 6.4 points improvement.). However, despite this paper citing AugMix multiple times, this table fails to include the results from the AugMix paper ([1]) on the same experiment. For most of the augmentations, AugMix [1] scores lower (better) than the baseline reported in this table (see Table 2 in [1]). Having omitted these comparisons, the evidence we can draw from Table 2 is limited.
  * Comparisons to traditional discretization methods are omitted. Overall, this method proposes to augment images using discretization. For the discretization, the “Vector-Quantized Variational AutoEncoder” is used. However, I miss a few intermediate steps. For example, a traditional auto encoder, as proposed in 2006 [2]. Moreover, how about traditional discretization techniques like JPEG compression? Given that JPEG compression has been around since 1992 [3], a comparison with this traditional method would increase the argument for the proposed method.

[1] https://arxiv.org/pdf/1912.02781.pdf
[2] Hinton and Salakhutdinov, 2006. “Reducing the dimensionality of data with neural networks”. Science.
[3] https://en.wikipedia.org/wiki/JPEG

# Statement

Recommendation: Reject
Reasons
  * The results miss comparisons to traditional techniques. For discretization, a “Vector-Quantized Variational AutoEncoder” is used, but the argument for not comparing to a traditional auto encoder, or even to a traditional compression technique like JPEG misses from the main text.
  * The empirical results are not comparable to other reports in literature. Table 1 compares on CIFAR-C with AugMix, but does not match results reported in that paper. Table 2 compares on ImageNet-C against a weak baseline, and omits the results on ImageNet-C for AugMix [1] and other publicly available methods [2].


[1] https://arxiv.org/pdf/1912.02781.pdf
[2] https://paperswithcode.com/sota/domain-generalization-on-imagenet-c


# Clarifying questions

  * The first paragraph of the introduction speaks of noisy images in “the real world”, the first paragraph of the Related Work speaks of “the real-world application of CNNs”, and the conclusion speaks of “real-world practice of deep learning”. However, the evaluations restrict themselves to ImageNet-C, which uses synthetic perturbations. There exist many more evaluations of robustness on the ImageNet label space: ImageNet-A [1], ImageNet-V2 [2], ObjectNet [3], and ImageNet-Video-Robust [4] . What is the reason for not including these evaluations when claiming “real world” robustness?
Finally, the text cites the texture/shape evaluation of Geirhos 2019 twice in the main text, but also that evaluation misses in the results section [5].
  * How does the Theorem 1 relate to your application of the auto encoder? From equation (2), I understand that $\hat{x}$ is a convex combination of the discretized image and the original image. The theorem states that the MI between input, x, and latent code, z, is upper bounded by K, but what does that tell us about $\hat{x}$?
  * The ablations in Table 5 explore the design choices for the proposed methods. However, in two cases, the ablation experiment scores better than the proposed method  (“DJMix w/o mixing”, 15.3; “DJMix w/ KL”, 15.3). What arguments are there for the design choices made in light of these numbers?

[1] Dan Hendrycks, Kevin Zhao, Steven Basart, Jacob Steinhardt, and Dawn Song. Natural adversarial examples. arXiv, abs/1907.07174, 2019
[2] Benjamin Recht, Rebecca Roelofs, Ludwig Schmidt, and Vaishaal Shankar. Do imagenet classifiers generalize to imagenet? In International Conference on Machine Learning, 2019.
[3]  Andrei Barbu, David Mayo, Julian Alverio, William Luo, Christopher Wang, Dan Gutfreund, Josh Tenenbaum, and Boris Katz. Objectnet: A large-scale bias-controlled dataset for pushing the limits of object recognition models. In Advances in Neural Information Processing Systems, 2019
[4] Vaishaal Shankar, Achal Dave, Rebecca Roelofs, Deva Ramanan, Benjamin Recht, and Ludwig Schmidt. A systematic framework for natural perturbations from videos. arXiv, abs/1906.02168, 2019
[5] Robert Geirhos, Claudio Michaelis, Felix A. Wichmann, Patricia Rubisch, Matthias Bethge, and Wieland Brendel. Imagenet-trained CNNs are biased towards texture; increasing shape bias improves accuracy and robustness. In ICLR, 2019


# Minor feedback

This minor feedback is not part of my assessment.

  * “Jansen–Shannon divergence” -> “Jensen–Shannon divergence”
  * The final paragraph of Section 3.1 and the final paragraph of Section 3.2 state the same text. Is this a mistake?
  * “models learn vulnerable representation” -> “models learn vulnerable representations”
  * “Pretrinaed” -> “pretrained”
  * “Degenerated” -> “perturbed”? (Section 4.0.2)
  * (Section 4.0.2) Reference to “Michaelis et al.” misses the year, 2019.
  * “Nonclassification” -> “non-classification” or “non classification”
  * The loss function in equation (1) is the same as the loss function in equation (1) of the AugMix paper. Even the choice for the JSD is the same. However, this connection is not mentioned in the main text. Given that you evaluate extensively against the AugMix results, please credit them appropriately for this design of the loss function.

---

> ### Author Response · Authors · 2020-11-25
> **Reply to R1**
>
> We appreciate your detailed comments and suggestions. In summary of our paper, we proposed a task-agnostic data augmentation method to improve the robustness of CNN models using mixing of images and their discretized ones. We are encouraged that R1 finds that our method is effective in a range of applications. We thank R1’s feedback, and we fixed the manuscript according to them. In the following, we are going to answer some of the points you mentioned.
>
> * Q1: Why not compare with traditional discretization methods and autoencoders? [reject reason1] A1: We compared our method with the standard autoencoder in the ablations (see Section 5.1) and showed the effectiveness of discretization. We did not include JPEG compression because it’s included in the benchmark corruptions.
>
> * Q2: Why the results of previous works in the tables are different from their papers? [reject reason2] A2: For a fair comparison, we reported the results of our experiments. Our environment is different from the authors of these previous works, and thus, the reported results are different. In this paper, we did not claim that our method is state of the art. Instead, we aim to introduce a novel approach that is robust in a wide range of applications. For this purpose, we believe that intensively comparing with prior methods is unnecessary.
>
> * Q3: Why the discretization autoencoder is not affected by the test-time image corruptions? A3: The discretization autoencoder is only used for training as data augmentation. Therefore, it is not affected by corruption.
>
> * Q4: What are the differences between the loss functions of AugMix and those of DJMix? A: We visually summarized the differences in Figure 4 in the appendix. Specifically, the design of task losses is different. Our choice is justified theoretically by Theorem 1 in Section 3.3 and empirically by the ablation experiments in Section 5.1.
>
> * Q5: What does Theorem 5 mean? A5: Theorem 5 states that DJMix makes CNNs learn more robust features by ignoring task-irrelevant details. See Section 3.3 again.

---

### Official Review · AnonReviewer1 · 2020-10-27
**A clever augmentation trick is used to keep object borders the same: this trick improves robustness to several common perturbations.**

**Rating:** 5
**Confidence:** 4

**Review:**


In this paper, the authors propose to linearly combine the following: the original image, and a discretized version thereof, to form an augmented image. Using the established pairing of a distance-based loss (for pushing logits from augmented/unaugmented data to be similar) and a classification loss, they show improved performance on a range of benchmarks that aim to test robustness to natural noise.

Overall, linearly combining an image with a discretized representation of itself is a neat trick to produce an augmentation where the contours and lines of an object are identical. There is a reasonable analysis of the choices made, and the method is tested in multiple image processing domains — showing broadly positive results. The authors frame the aim of the work as needing 'robust methods that are agnostic to test-time noise a-priori'. While this framing is arguably unmet, the motivation is clear and interesting.

Taken as a whole, the lack of space dedicated to analyzing the results is detrimental to the work. There are 5 separate tables in the main work, each of which only receives minimal discussion in the text. While simplistic conclusions are drawn from the tables ("table 4 shows ... better performance on almost all corruption type") examples that don't adhere to the general trend are ignored — why is the 'pixel' case in table 4 better for the baseline, for instance? Also, there is no indication of any sort of variance of the results — while this can be resource-intensive, it would be welcomed at least for CIFAR datasets. I would recommend more discussion of results and a more thorough examination of the trends in the results. To make space for this, I would recommend reducing §4. The methods are fairly well-explained before this, and multiple pages are spent on details that could safely be moved to the appendix.

There is also a lack of theoretical analysis of this method. For instance, the task loss is different to the augmix baseline — but this is not really explored except by experiment. Are there reasons why the different loss might be theorized to improve results?

While Table 3 and 4 show results against an augmix without geometric augmentations (which is reasonable), I would recommend some discussion of how you COULD use geometric transforms for these methods. For instance, could you do some sort of mapping between pixels for the consistency loss (given by the generating transform of the augmentation)?

In general, the robustness of the proposed method is slightly overstated. Simply performing better on some of the corrupted datasets (and not all of them) is not enough for the sort of guarantees that would matter for truly safety-critical operations. This is especially true when the corruptions are applied one-at-a-time. I feel that the paper should at least acknowledge work on provable guarantees of robustness and explain why that type of analysis is irrelevant here, or talk about failings of previous models that this work highlights.

*minor points:*
* Throughout the manuscript, the authors refer to the fact that the augmented images are 'nearly impossible to distinguish' from the un-augmented images. Does this hold for humans and machines both? How well can you train a simple classifier to spot this?
* I recommend a simplified explanation of the effect on segmentation — something like "because the proposed augmentation does not 'move' the image at all, the edges are in the same place, leaving the labels untouched.
* 'we omitted five operations' — please list all of the omitted operations in the main text.
* 'ignore malicious features' — I would choose a different word than 'malicious'.
* 'for the real-world application of CNNs...' I would contend that CNNs are already very-much used in the real world!

---

> ### Author Response · Authors · 2020-11-25
> **Reply to R2**
>
> We appreciate your detailed comments and suggestions. In summary of our paper, we proposed a task-agnostic data augmentation method to improve the robustness of CNN models using mixing of images and their discretized ones. We are encouraged that R2 finds our method is neat, the analysis is reasonable, and the results are broadly positive. We thank R2’s kind suggestions. We fixed the following points.
>
> - We listed five omitted operations (defocus blur, glass blur, motion blur, zoom blur, and gaussian blur).
> - We used “harmful features” instead of “malicious features”.
> - We fixed the values in Table 2,3 (a script to create tables was wrong) and enhanced the discussion of the results.
> - We added standard deviations in Table 5.
>
> In the following, we are going to answer some of the points you mentioned.
>
> 1. Q: Why DJMix uses a different task loss from AugMix? A: As far as we understand, the task loss of AugMix is not theoretically justified. Our task loss design is theoretically motivated (See Theorem 1) and has the empirical improvement of the score (See Table 5b).

---

### Official Review · AnonReviewer3 · 2020-10-30
**Simple and effective idea, but need more experiments**

**Rating:** 5
**Confidence:** 4

**Review:**

#### Summary

This paper proposes an image augmentation strategy DJMix for helping the network robustness in image recognition tasks over the corrupted images. The experiments show it performs better than AugMix on segmentation and detection over pascal dataset.

#### Novelty

The method is simple while effective as demonstrated with experiments, though it is similiar with Augmix. The idea could related to some robust training methods proposed. for example, Improving Adversarial Robustness by Data-Specific Discretization, which should also be discussed.

#### Writing

Writing is easy to follow， but the figure is comparably small (For example, images in Fig.1, 3), the author might need to adjust the size a bit.


#### Experiments

The classification results seem to be limited，In classification, it seems the results are comparably worse than mixup (CIFAR10) and for segmentation, the baseline results is only 76.8 which is 12 percent lower than current SoTA algorithms. It would be more strong by comparing with these networks.

For more experiments, I recommended experiments over larger dataset such as Coco detection and segmentation tasks, in addition, it could be better comparing with other augmentation strategy such as AutoAug and neural architecture searched.

page 5: pretrinaed -> "pretrained"

---

> ### Author Response · Authors · 2020-11-25
> **Reply to R3**
>
> We appreciate your detailed comments and suggestions. In summary of our paper, we proposed a task-agnostic data augmentation method to improve the robustness of CNN models using mixing of images and their discretized ones. R3 finds our method is simple but effective and our manuscript is easy to follow. We thank R3 discovers our typo. We fixed this. We also adjusted the size of the figures. Below, we address the comments in your reviews.
> * Q1: The results are worse than state-of-the-art methods. A1: In this paper, we don’t introduce a state-of-the-art method, but a novel method with a task-agnostic property, and, importantly, the scores are close to the task-specific robust method (AugMix) in the classification task.
> * Q2: Chen+18 “Improving Adversarial Robustness by Data-Specific Discretization” should be related. A1: Thank you for introducing an interesting paper. Though both Chen+18 and our paper use discretization, Chen+18 used discretization for preprocessing for training and testing, whereas our paper proposed to use discretization for data augmentation (training-time only).

---

### Official Review · AnonReviewer4 · 2020-10-30
**This paper proposes DJMix, a data augmentation method to improve the robustness by mixing each training image and its discretized one. Discretization is done in an autoencoder and the mixed images are hard to distinguish from the original images, making it applicable to various image recognition tasks.**

**Rating:** 4
**Confidence:** 4

**Review:**

This paper proposes DJMix, a data augmentation method to improve the robustness by mixing each training image and its discretized one, and analyzes DJMix theoretically from the Information Bottleneck perspective. In addition, it creates datasets, Segmentation-C and Detection-C, to benchmark the robustness of CNN models on semantic segmentation and detection tasks.

Although it achieves good results in some settings, e.g. CIFAR 10 in Table 1, when combined with Random Augmentation (RA), it looks like the overall results are just comparable even slightly worse than AugMix. In addition, the paper may also need to present results with RA only as good results are almost all achieved by DJMix + RA.  Knowing the sole effect of RA is also important. In addition, the results on ImageNet in Table 2 don't compare with other methods, e.g. AugMix, making it weaker. Results of AugMix in Table 3 and 4 don't use geometric transformations doesn't make sense to me as it can achieve good results according to Table 1. Comparisons with other methods are also not enough in Semantic segmentation and detection tasks.

Given the aforementioned evidence, I think it should add more experiments to support its argument. Therefore, I prefer to reject this paper until it has more solid experiments.

---

> ### Author Response · Authors · 2020-11-25
> **Reply to R4**
>
> We appreciate your detailed comments and suggestions. In summary of our paper, we proposed a task-agnostic data augmentation method to improve the robustness of CNN models using mixing of images and their discretized ones. We are encouraged that R4 acknowledge that we created datasets for this research and we analyzed the proposal theoretically. Below, we address the comments in your reviews.
> As of scores, we did not claim our method is superior to AugMix. We proposed a task-agnostic method, and, importantly, the scores are close to the task-specific baseline (AugMix). We used RA without geometric transformations on detection and segmentation, because "the mixed images of geometric transformations cannot have trivial targets in non-classification tasks, for instance, semantic segmentation and detection”, as written on page 1. We emphasized this in the section on experiments.

---

### Decision · Program_Chairs · 2021-01-07
**Final Decision**

**Decision:**

Reject

**Comment:**

This paper proposes to improve the robustness of computer vision models through a new augmentation strategy. There are two primary contributions of the work, first the use of a bottleneck autoencoder to generate discretized variants of the clean image, and second a slight variant of the task loss, where the task loss is evaluated on the augmented image vs the clean image as is done in prior work. Reviewers argued that the method did not meaningfully improve upon prior work, the method alone underperforms AugMix on existing benchmarks, and when combined with some additional augmentations from AugMix the gains were marginal. Additionally, when there were gains in robustness it was unclear as to the source. The work would be improved with additional experimental evidence that the claimed benefit of the information bottleneck is substantial for improving robustness (for example, when DJMix+RA outperforms AugMix, is this due to the use the of autoencoder or is it primarily due to the new task loss?). I recommend the authors incorporate additional reviewer feedback and resubmit.